

SciPost Phys. Lect. Notes 37 (2022)

# Dark matter in astrophysics/cosmology

**Anne M. Green**[⋆]

School of Physics and Astronomy, University of Nottingham,
University Park, Nottingham, NG7 2RD, UK

⋆ anne.green@nottingham.ac.uk

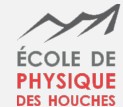

*Part of the Dark Matter*
*Session 118 of the Les Houches School, July 2021*
*published in the Les Houches Lecture Notes Series*

## Abstract

These lecture notes aim to provide an introduction to dark matter from the perspective of astrophysics/cosmology. We start with a rapid overview of cosmology, including the evolution of the Universe, its thermal history and structure formation. Then we look at the observational evidence for dark matter, from observations of galaxies, galaxy clusters, the anisotropies in the cosmic microwave background radiation and large scale structure. To detect dark matter we need to know how it's distributed, in particular in the Milky Way, so next we overview relevant results from numerical simulations and observations. Finally, we conclude by looking at what astrophysical and cosmological observations can tell us about the nature of dark matter, focusing on two particular cases: warm and self-interacting dark matter.

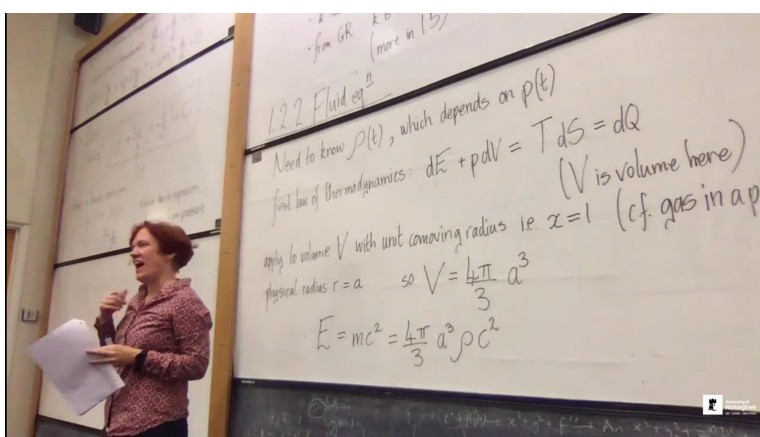

# 0   Introduction

These lectures aim to provide the participants in the 'Les Houches Summer School 2021: Dark Matter' with the background knowledge of cosmology/astrophysics required for the other courses during the School, and also for carrying out research in the field of dark matter (DM). We start, in Sec. 1, with a brief introduction to cosmology, at the level of an undergraduate/bachelors course, focusing on the topics that are most relevant for DM. This will be a rapid 'crash course' for people who haven't already studied such material and a recap for those who have. In Sec. 2 we move on to the observational evidence for cold, non-baryonic dark matter, on scales ranging from individual galaxies to the Universe as a whole. Understanding the distribution of DM, in particular within the Milky Way, is crucial for its detection, so this is the focus of Sec. 3. Finally, in Sec. 4 we look at what observations can tell us about the nature of DM, focusing on warm and self-interacting DM.

The introduction to each section includes recommendations of books and review papers for further reading. For 'background material' I tend to reference books/reviews papers, partly because they're typically more accessible than the original papers, and partly because tracking down the appropriate original references across such a wide range of topics would take more time than is feasible. For more specific topics I've attempted to cite review papers and in some cases the first and/or most relevant recent papers, as 'jumping-off points' into the literature.[1] In cases where there are many other sources which could have equivalently been cited, this is indicated using "see e.g. Ref. [X]". One section of the notes doesn't correspond to one lecture. In particular Sec. 1, 'A brief introduction to cosmology', is significantly longer than the subsequent sections. The depth of coverage is also variable; I've tried to explain key concepts in detail, but there are other, incidental, things which I mention briefly, so that if you encounter them you know roughly what they are. In some cases I'll introduce topics which are covered in more detail later in the School by experts on the topic.[2]

# 1   A brief introduction to cosmology

## 1.0   Introduction

This Section aims to provide a concise overview of the aspects of cosmology that are relevant for understanding the evidence for, production of and distribution of DM. After an introduction to the key equations and quantities involved (Sec. 1.1), we'll start by studying the evolution of the Universe (Sec. 1.2). We'll then move on to its thermal history (Sec. 1.3), with particular emphasis on two epochs that are important in the evidence for DM: nucleosynthesis (Sec. 1.3.1) and the cosmic microwave background (Sec. 1.3.2). Next we look at two topics that play a key role in probing DM: structure formation (Sec. 1.4) and gravitational lensing (Sec. 1.5). Then we very briefly study inflation (Sec. 1.6), a proposed period of accelerated expansion in the early Universe. While not directly relevant to DM, it provides a mechanism for producing the initial fluctuations from which structures subsequently form, and is also relevant to the production of some DM candidates (for instance Primordial Black Holes, see lectures by Bernard Carr and Florian Kühnel in this volume). Finally we look briefly at Type 1a supernovae and the evidence for the recent accelerated expansion of the Universe. This is a key piece of evidence for the ΛCDM cosmological 'standard model', and the accelerated expansion of the universe at late times affects the growth of density perturbations and hence

---

[1]How successful I've been in doing this appropriately probably decreases the further I get from my own research interests.

[2]In the case of oversimplifications/discrepancies, trust the expert.

the formation of structures on large scales.

It would usually take 20+ hours of lectures to cover this material, and significantly longer if the key equations are derived 'from first principles' using general relativity. Therefore our treatment will be necessarily superficial (and unsatisfactory for fans of rigour).

**Recommended further reading**

My favourite bachelors level textbook is Ryden's 'Introduction to cosmology' [1]. If you'd like to go into more detail, then Tong's online lecture notes [2] are fantastic, while 'Volume III: Galaxies and Cosmology' of Padmanabhan's series of Theoretical Astrophysics textbooks [3] and Dodelson and Schmidt's 'Modern cosmology' [4] are useful for structure formation. The 'Particle Data Group Review of Particle Physics' [5] contains relatively concise overviews of Big Bang Cosmology (Olive & Peacock), Nucleosynthesis (Fields, Molaro & Sarkar) and the Cosmic Microwave Background (Scott & Smoot).

## 1.1 Key equations and quantities

We'll start with a concise overview of the key equations and quantities for describing the evolution of the universe.[3] The Friedmann equation tells us how the expansion of the universe (via the Hubble parameter, $H$) depends on its contents (through their energy density, $\epsilon$, and the cosmological constant, $\Lambda$) and geometry (via the curvature parameter, $k$):

$$H^2 \equiv \left(\frac{\dot{a}}{a}\right)^2 = \frac{8\pi G}{3}\epsilon - \frac{k}{a^2} + \frac{\Lambda}{3}, \tag{1}$$

where $G$ is Newton's gravitational constant, the scale factor $a(t)$ parameterises the expansion of the universe, and is often (but not always) normalised to unity today, dots denote derivatives with respect to time, and we have followed the convention of setting the speed of light, $c$, equal to 1 ($c = 1$). In cosmology subscript zero is sometimes used to denote the value of a variable at the present day. For instance, the Hubble constant, $H_0$, is the present day value of the Hubble parameter. It is often written in the form $H_0 = 100h\,\mathrm{km\,s^{-1}\,Mpc^{-1}}$, with the dimensionless constant $h$ parameterising the uncertainty. We'll see in Sec. 2 that observations often constrain combinations of parameters which include $h$, so the uncertainty in the present day expansion rate of the Universe propagates into uncertainties in the densities of the various components of the Universe. Planck measurements of the Cosmic Microwave Background (CMB), see Sec. 1.3.2, plus other 'early Universe' probes, find (roughly) $h \approx 0.674 \pm 0.005$, while local 'distance ladder measurements' find $h \approx 0.73 \pm 0.02$. The discrepancy between these measurements, known as the 'Hubble tension', is currently one of the hottest topics in cosmology. For further details see e.g. Ref. [6].

The fluid equation describes how the density of any component of the universe varies with time due to its expansion:

$$\dot{\epsilon} = -3H(\epsilon + p), \tag{2}$$

where $p$ is pressure. When studying the evolution of the universe its constituents can be grouped together, by their equation of state $p = w\epsilon$, where $w$ is the equation of state parameter. Radiation is relativistic particles (i.e. photons and light neutrinos), which have $w = 1/3$, while matter is non-relativistic particles (i.e. baryons[4] and cold dark matter) which have $w = 0$. A

---

[3]I've tried to use 'universe' when referring to a generic, theoretical universe, and 'Universe' when referring to the actual Universe that we live in, however the distinction isn't always clear-cut.

[4]Cosmologists often use 'baryons' to mean both baryons and leptons. Since the electron mass is much smaller than that of the proton and neutron, the contribution to the energy density of leptons is negligible compared to that of baryons.

cosmological constant can alternatively be described as a fluid with $w = -1$, and any dominant fluid with $w < -1/3$ ('dark energy') will cause accelerated expansion (Sec. 1.7).

The critical density, $\epsilon_c$, is the (time-dependent) value of the energy density for which the geometry of the universe is flat ($k = 0$):

$$\epsilon_c = \frac{3H^2}{8\pi G}. \tag{3}$$

The density parameter denotes the density (either total or of a specific component) relative to the critical density: $\Omega = \epsilon/\epsilon_c$. As we'll see in Sec. 2.3, observations of the anisotropies in the CMB tell us that the geometry of the Universe is very close to flat: $k \approx 0$ and $\Omega_{\text{total}} \approx 1$.[5]

It's often convenient to use comoving coordinates that are 'carried along' with the expansion of the universe: $\mathbf{R}(t) = a(t)\mathbf{x}$, where $\mathbf{R}(t)$ is the usual physical coordinate and $\mathbf{x}$ is the comoving coordinate.

A homogeneous, isotropic, expanding/contracting universe is described by the Friedmann-Lemaitre-Robertson-Walker metric:

$$\mathrm{d}s^2 = -\mathrm{d}t^2 + a^2(t)\left[\frac{\mathrm{d}r^2}{1-kr^2} + r^2(\mathrm{d}\theta^2 + \sin^2\theta\,\mathrm{d}\phi^2)\right], \tag{4}$$

where $\mathrm{d}s^2$ is the separation between two events in spacetime and $(r, \phi, \theta)$ are comoving spherical coordinates. Photons travel on null-geodesics with $\mathrm{d}s^2 = 0$, and (in a flat $k = 0$ universe) have

$$\mathrm{d}r = \frac{\mathrm{d}t}{a(t)}. \tag{5}$$

The expansion of the universe leads to cosmological redshift of the wavelength of photons: $\lambda \propto a$. The redshift, $z$, is related to the scale factor by $a = 1/(1+z)$. Since redshift is measurable (via the change in wavelength of spectral lines) it's often more practical than $a$ or $t$ for parameterising the expansion of the universe. Due to the finite speed of light, photons can only travel a finite distance, known as the horizon distance, in any finite time:

$$d_{\text{hor}} = a(t)\,r_{\text{hor}} = a(t)\int_0^{r_{\text{hor}}}\mathrm{d}r = a(t)\int_0^t \frac{\mathrm{d}\tilde{t}}{a(\tilde{t})} \sim H^{-1}, \tag{6}$$

where $r_{\text{hor}}$ is the comoving horizon distance. The pre-factor in front of the $H^{-1}$ in the final expression depends on the form of $a(t)$, which (as we'll see in the next section) depends on the contents of the universe.

## 1.2 Evolution of the Universe

As we'll see at the end of this subsection, for significant periods of time the density of the Universe is dominated by a single component. Therefore it's instructive to initially consider the evolution of simple, single component, flat universes.

Matter has equation of state $p_m = 0$ and hence from the fluid equation, Eq. (2), its energy density decreases as $\epsilon_m \propto a^{-3}$. Physically this is because number density is inversely proportional to volume. Inserting the scale factor dependence of the energy density into the Friedmann equation, Eq. (1), and integrating shows that a flat ($k = 0$) matter dominated universe expands as $a \propto t^{2/3}$. If the universe is dominated by a cosmological constant, $\Lambda$, then, by integrating the Friedmann equation, Eq. (1), $a(t) \propto \exp(\sqrt{\Lambda/3}\,t)$.

Radiation has equation of state $p_r = \epsilon_r/3$ and in this case $\epsilon_r \propto a^{-4}$. The additional factor of $a$ relative to matter comes from the decrease in the energy of individual relativistic particles,

---

[5]This is a good thing; various calculations are much simpler for a flat universe than for open or closed universes.

due to redshift. A flat radiation dominated universe expands as $a \propto t^{1/2}$. Photons are bosons and have a blackbody distribution, so that the energy density in the frequency interval $f$ to $f + \mathrm{d}f$ is given by

$$\epsilon(f)\mathrm{d}f = 8\pi h \frac{f^3 \, \mathrm{d}f}{\exp\left(\frac{hf}{k_\mathrm{B}T}\right) - 1}, \tag{7}$$

here, $h$ is the Planck constant, $k_\mathrm{B}$ the Boltzmann constant and $T$ is temperature. The mean photon energy is $2.7k_\mathrm{B}T$, and there is a long tail of high $E$ photons (which has consequences for the thermal history of the Universe, see Sec. 1.3). From integrating Eq. (7) the total photon energy density is $\epsilon_\gamma = \alpha T^4$, where $\alpha = 7.6 \times 10^{-16} \, \mathrm{J \, m^{-3} \, K^{-4}}$. Using the present day CMB temperature, $T_0 = 2.725 \, \mathrm{K}$ [7], the present day photon number density is $n_{\gamma,0} = 4.1 \times 10^8 \, \mathrm{m^{-3}}$, which is roughly nine orders of magnitude greater than the present day baryon number density. Again, the fact that there is a huge number of photons for every baryon has consequences for the thermal history of the Universe. The neutrino energy density is related to the photon energy density by

$$\epsilon_\nu = \left[ 3 \times \frac{7}{8} \times \left( \frac{4}{11} \right)^{4/3} \right] \epsilon_\gamma = 0.68\epsilon_\gamma, \tag{8}$$

where the numerical factors in the middle expression come respectively from the fact that i) there are 3 species of light neutrino, ii) neutrinos are fermions rather than bosons and iii) electron-positron annihilation produces, and increases the temperature of, photons but not neutrinos. In general

$$\epsilon_\mathrm{r} = \frac{\pi^2}{30} g_\star T^4, \tag{9}$$

where $g_\star$ is the (temperature/time dependent) total number of effectively massless degrees of freedom. Combining this expression with $\epsilon_\mathrm{r} \propto a^{-4}$, gives $a \propto g_\star^{-1/4} T^{-1}$, so that when $g_\star$ is constant $a \propto T^{-1}$. Finally, there's a useful 'rule of thumb' (which can be derived by either using the Friedmann equation or the scalings of $a$, $t$ and $T$) that during radiation domination, time and temperature are related by

$$\left( \frac{1\,\mathrm{s}}{t} \right)^{1/2} \approx \left( \frac{k_\mathrm{B}T}{1\,\mathrm{MeV}} \right). \tag{10}$$

Finally, to finish the evolution of the universe, we'll rewrite the Friedmann equation, Eq. (1), for a general universe containing matter, radiation and a cosmological constant, in a more useful form. Using the expressions we've previously seen for the evolution of the energy density of matter and radiation, $\epsilon_\mathrm{m} \propto a^{-3}$ and $\epsilon_\mathrm{r} \propto a^{-4}$, the relationship between scale-factor and redshift, $a = 1/(1+z)$, the expression for the critical density, $\epsilon_\mathrm{c}$, Eq.(3), and introducing definitions for the curvature and cosmological constant density parameters, $\Omega_k = -k/H^2$ and $\Omega_\Lambda = -\Lambda/H^2$, gives

$$H^2 = H_0^2 \left[ \Omega_{\mathrm{r},0}(1+z)^4 + \Omega_{\mathrm{m},0}(1+z)^3 + \Omega_{\mathrm{k},0}(1+z)^2 + \Omega_{\Lambda,0} \right]. \tag{11}$$

We've ordered the terms here according to how rapidly they decrease as the universe expands, starting with radiation, which falls off the most rapidly. This equation can be solved numerically given (observationally determined) values for the present day Hubble parameter and density parameters. However, to a good approximation the density components each dominate in turn i.e. the universe starts off radiation dominated, becomes matter dominated and then at late times is dominated by the cosmological constant. In principle the curvature could dominate in between matter and the cosmological constant. However, as we'll see in Sec. 2.3.2, we live in a universe which is very close to flat, the curvature density is hence small and it never dominates. Fig. 1 shows the evolution of the total density, and the contributions from radiation, matter, and a cosmological constant, with the scale factor.

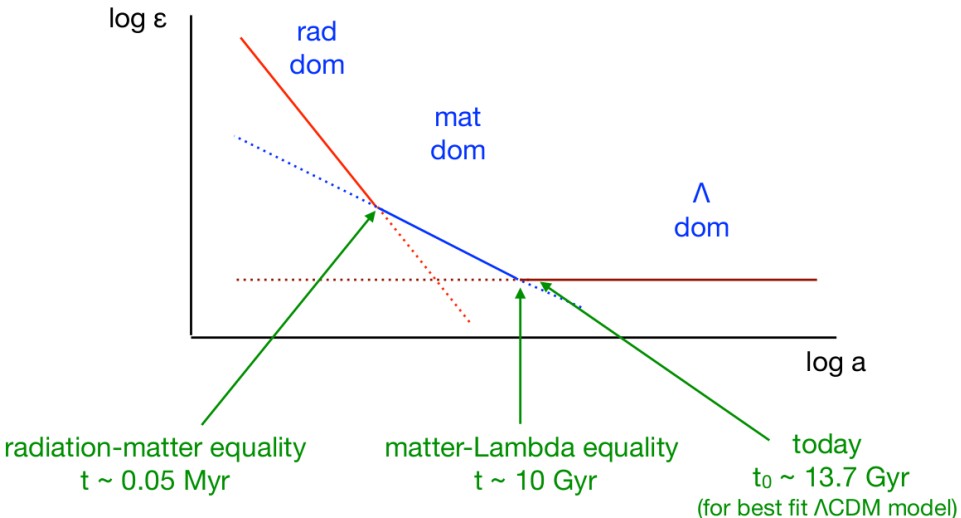

n.b. slopes not to scale

Figure 1: Evolution of the log of the density of the universe, $\epsilon$, with the log of the scale factor, $a$. The universe starts off radiation dominated (red line), becomes matter dominated (blue) and then at late times is dominated by the cosmological constant (burgundy).

## 1.3 Thermal history

### 1.3.1 Nucleosynthesis

(Big Bang) Nucleosynthesis is the synthesis of the nuclei of the light elements (Deuterium, D, Helium-3, $^3$He, Helium-4, $^4$He, and Lithium-7, $^7$Li) when the Universe was seconds to minutes old. For a more detailed overview see the Nucleosynthesis section (Fields, Molaro & Sarkar) of the 'Particle Data Group review of particle physics' [5].

Before $t \sim 1\,\mathrm{s}$ ($k_B T \sim 1\,\mathrm{MeV}$) protons and neutrinos are kept in thermal equilibrium by weak interactions:

$$\mathrm{n} + \nu_e \quad \leftrightarrow \quad \mathrm{p} + \mathrm{e}^-, \tag{12}$$

$$\mathrm{n} + \mathrm{e}^+ \quad \leftrightarrow \quad \mathrm{p} + \bar{\nu}_e. \tag{13}$$

The neutrons and protons are non-relativistic ($k_B T \ll m_p$), so they have a Maxwell-Boltzmann distribution and their relative number densities are given by

$$\frac{N_n}{N_p} = \left(\frac{m_n}{m_p}\right)^{3/2} \frac{\exp\left[-m_n/(k_B T)\right]}{\exp\left[-m_p/(k_B T)\right]} \approx \exp\left[-\frac{(m_n - m_p)}{k_B T}\right]. \tag{14}$$

While $k_B T \gg (m_n - m_p) = 1.3\,\mathrm{MeV}$, $N_n \sim N_p$, however once the thermal energy drops below the difference in the rest mass energies $N_n < N_p$. A full calculation shows that once $k_B T_{\mathrm{fo}} \sim 0.8\,\mathrm{MeV}$, the timescale on which the weak reactions occur is longer than the age of the Universe, and therefore interconversion of neutrons and protons ceases (or 'freezes-out'). At this point $N_n/N_p \approx 0.2$.

The subsequent production of the nuclei of the light elements occurs via a chain of reactions:

$$p + n \quad \rightarrow \quad D, \tag{15}$$

$$D + p \quad \rightarrow \quad {}^3He, \tag{16}$$

$$D + D \quad \rightarrow \quad {}^4He, \dots \tag{17}$$

Initially the high energy tail of the photon distribution can destroy nuclei, however this stops once the thermal energy drops to $k_B T_{nuc} \sim 0.1 \, \mathrm{MeV}$. Between $T_{fo}$ and $T_{nuc}$ free neutrons decay into protons, and the neutron to proton ratio drops to $N_n / N_p \approx 0.18$. Below $T_{nuc}$, the majority of the remaining neutrons form ${}^4He$, the most stable light nucleus, with trace amounts of the heavier nuclei being formed. The mass fractions of each isotope are: $Y_{{}^4He} = 0.23 - 0.24$, $Y_D \approx 10^{-4}$, $Y_{{}^3He} = 10^{-5}$, $Y_{{}^7Li} = 10^{-10}$. The exact abundances depend on the photon to baryon ratio, or equivalently (since the photon number density is known from the CMB temperature) the abundance of baryons. Therefore by comparing the theoretical predictions with observations, in particular of the Deuterium abundance from absorption of light from quasars by primordial gas clouds, the baryon density parameter can be determined: $0.021 \leq \Omega_b h^2 \leq 0.024$.[6] This is consistent with, but less precise, than the determination from the anisotropies in the CMB, which we'll cover in Sec. 2.3.

### 1.3.2 Cosmic microwave background

The ionisation energy of hydrogen is $E_{ion} = 13.6 \, \mathrm{eV}$. At high temperatures photons have energy greater than this and rapidly ionise any atoms that form. Therefore at early times the Universe is composed of nuclei and electrons, and photons scatter frequently off these charged particles. As the Universe expands and cools the energy of the photons drops and atoms form. This process is known as recombination and occurs at $t \sim 0.25 \, \mathrm{Myr}$ when $k_B T \sim 0.32 \, \mathrm{eV}$ (this is less than $13.6 \, \mathrm{eV}$ since $n_\gamma \gg n_b$). The Universe becomes neutral and photons stop scattering and subsequently free-stream through the Universe. This process is known as decoupling and occurs at $t \sim 0.37 \, \mathrm{Myr}$ when $k_B T \sim 0.26 \, \mathrm{eV}$. The resulting Cosmic Microwave Background has a black body spectrum with present day temperature $T_0 = 2.7255 \pm 0.0006 \, \mathrm{K}$ [7].

The initial small density fluctuations from which structures form (we'll study this process in more detail in Sec. 1.4) lead to fluctuations in the temperature of the CMB. The fluctuation distribution depends on the initial primordial perturbations, and also on the contents of the Universe (because they affect the growth of fluctuations, and also because of the projection of length scales into angles on the sky). We'll look at how CMB observations constrain these quantities in Sec. 2.3.

The temperature fluctuations are analysed by expanding in spherical harmonics, $Y_l^m(\theta, \phi)$:

$$\frac{\Delta T(\theta, \phi)}{\bar{T}} \equiv \frac{T(\theta, \phi) - \bar{T}}{\bar{T}} = \sum_{l=1}^{\infty} \sum_{m=-l}^{l} a_{lm} Y_l^m(\theta, \phi), \tag{18}$$

where $\bar{T}$ is the average temperature, $T(\theta, \phi)$ the temperature in a particular direction and $a_{lm}$ the coefficients of the expansion. The angular power spectrum, $C_l$, is calculated by taking the statistical average of the coefficients

$$C_l = \langle |a_{lm}|^2 \rangle. \tag{19}$$

Small values of the multipole moment, $l$, correspond, roughly, to large angular separations and vice versa. There are several characteristic regions (see Fig. 2):

---

[6]Values of the density parameters are almost always quoted at the present day, therefore here, and subsequently, we follow the convention of dropping the usual subscript '0' explicitly denoting this.

- The 'Sachs-Wolfe' plateau at low $l$. In this regime the temperature variations arise from variations in the gravitational potential.

- The acoustic (or Doppler) peaks at intermediate $l$. These come from oscillations in the photon-baryon fluid due to the competition between gravity and pressure (which arises due to interactions between photons and electrons).

- The Silk damping tail at high $l$. Due to diffusion of photons during the recombination process the temperature fluctuations on small scales are damped.

Additional information can be obtained from the polarization and lensing of the CMB photons. $E$ mode polarization is due to Thomson scattering of the CMB photons off free electrons, while $B$ mode polarization can arise from lensing of $E$ modes, dust, or primordial tensor perturbations. Photons are deflected by gravitational potentials, and this smooths out the acoustic peaks.

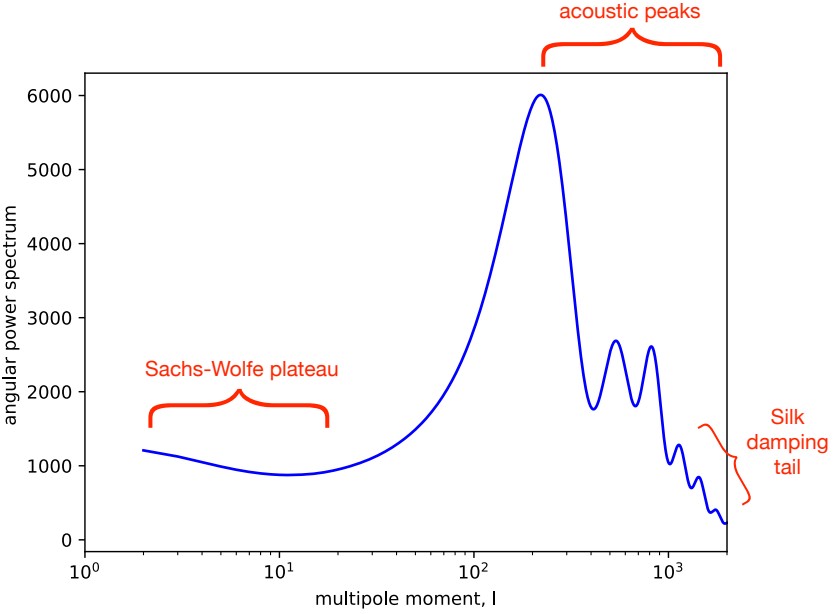

Figure 2: The theoretical CMB angular power spectrum, $T_0^2 l(l+1)C_l/(2\pi)$ in $(\mu\,\mathrm{K})^2$, as a function of multipole moment, $l$, for the best fit $\Lambda$CDM model, calculated using the CAMB Web Interface [8].

## 1.4 Structure formation

Structures (galaxies, galaxy clusters) form via gravitational instability: small initial overdensities grow due to gravity. How the perturbations grow, and hence how galaxies cluster, depends on the contents of the Universe. Perturbations on a given scale can only grow once that scale is smaller than the horizon scale, Eq. (6). A comoving scale, $k$, is said to 'enter the horizon' when its inverse, $(k^{-1})$, is equal to the comoving Horizon scale, $(H^{-1}/a)$, i.e. $k = aH$. What follows is a somewhat qualitative outline of how perturbations evolve (c.f. Ref. [1]) for a more rigorous analysis see e.g. Refs. [2, 4].

We'll first consider an over-dense sphere in a pressure-less static universe with mass density $\rho = \bar{\rho}(1+\delta)$, where $\delta = (\rho - \bar{\rho})/\bar{\rho}$ is the density perturbation. From considering the gravitational acceleration at the sphere's surface, and using mass conservation, $\delta$ evolves according to $\ddot{\delta} = 4\pi G \bar{\rho} \delta$ and hence the perturbation grows exponentially with characteristic timescale $t_{\mathrm{dyn}} = 1/(4\pi G\bar{\rho})^{1/2}$. Non-zero pressure will resist collapse, but this takes a time $t_{\mathrm{pre}} \sim R/c_{\mathrm{s}}$ where $R$ is the size of the perturbation and $c_{\mathrm{s}} = (\mathrm{d}p/\mathrm{d}\epsilon)^{1/2} = \sqrt{w}$ is the sound speed of the medium. Perturbations will grow if $t_{\mathrm{pre}} > t_{\mathrm{dyn}}$ which is the case if $R$ is greater than the Jeans length, $\lambda_J \sim c_{\mathrm{s}} t_{\mathrm{dyn}} \sim c_{\mathrm{s}}/(G\bar{\rho})^{1/2}$.

In an expanding universe, on sub-horizon scales DM perturbations evolve according to

$$\ddot{\delta} + 2H\dot{\delta} - \frac{3}{2}\Omega_{\mathrm{m}}H^2\delta = 0\,. \tag{20}$$

During radiation domination ($\Omega_{\mathrm{m}} \ll 1$ and $a \propto t^{1/2}$): $\delta(t) = B_1 + B_2 \ln t$, while during matter domination, ($\Omega_{\mathrm{m}} \approx 1$ and $a \propto t^{2/3}$): $\delta(t) = D_1 t^{2/3} + D_2 t^{-1}$, where $B_{1/2}$ and $D_{1/2}$ are constants. Hence DM perturbations grow as a power-law from radiation-matter equality. However, before decoupling baryons are tightly coupled to photons and have $c_{\mathrm{s}} = 1/\sqrt{3}$, and hence baryonic perturbations can't grow. After decoupling baryons 'fall into the potential wells' created by DM. We'll see in Sec. 2.3.1 that these effects lead to evidence for DM from the amplitude of the temperature anisotropies in the CMB.

The nature of DM affects the evolution of the density perturbations. Cold dark matter (CDM) decouples when non-relativistic, and has very small velocity dispersion today. Hot dark matter (HDM) decouples when relativistic and has non-zero velocity dispersion today. We'll look at the intermediate case of warm dark matter in Sec. 4.1. HDM with mass $m_{\mathrm{hdm}}$ becomes non-relativistic at temperature $T_{\mathrm{hdm}}$, when $3k_B T_{\mathrm{hdm}} \approx m_{\mathrm{hdm}}$, prior to this it free streams and erases density perturbations on length scales smaller than $d_{\mathrm{min,hdm}} \approx t_{\mathrm{hdm}}$. For $m_{\mathrm{hdm}} > 2.4\,\mathrm{eV}$ (so that this happens before radiation-matter equality), this corresponds to a mass

$$M_{\mathrm{min,hdm}} \approx 10^{16} \left(\frac{m_{\mathrm{hdm}}}{3\,\mathrm{eV}}\right)^{-3} M_\odot\,, \tag{21}$$

i.e. with HDM, perturbations on scales smaller than galaxy clusters are erased, and structure formation is 'top down': large objects form first. Erasure of perturbations also occurs for (thermal relic) CDM but, because of its much smaller velocity dispersion, on much smaller scales. The first WIMP halos to form are roughly Earth mass microhalos, $M_{\mathrm{min,cdm}} \sim 10^{-6} M_\odot$, [9], with larger halos forming from mergers and accretion.

The amplitude of the density perturbations is quantified via the power spectrum $P(k) = \langle|\delta_k|^2\rangle$, where $\delta_k$ is the Fourier transform of the density perturbation. The power spectrum can be written in the form

$$P(k,t) = \frac{2\pi^2}{k^3} T^2(k,t)\mathcal{P}(k)\,, \tag{22}$$

where $T(k,t)$ is the transfer function which describes the evolution of the density perturbations and $\mathcal{P}(k)$ is the primordial power spectrum, which is usually assumed (on cosmological scales) to have the form

$$\mathcal{P}(k) = A_{\mathrm{s}}\left(\frac{k}{k_\star}\right)^{n_{\mathrm{s}}-1}\,, \tag{23}$$

where $k_\star$ is a fiducial scale (which is usually taken to be roughly in the centre of the range of scales probed by the data), $A_{\mathrm{s}}$ is the amplitude and $n_{\mathrm{s}}$ is the scalar spectral index. A scale-invariant power spectrum (same amplitude on all scales) has $n_{\mathrm{s}} = 1$, while slow-roll inflation models (see Sec. 1.6) produce perturbations with $n_{\mathrm{s}}$ close to 1. From Planck CMB temperature anisotropy measurements, $n_{\mathrm{s}} = 0.959 \pm 0.006$ and $A_{\mathrm{s}} = 2.2 \times 10^{-9}$ for $k_\star = 0.05\,\mathrm{Mpc}^{-1}$ [10].

The typical amplitude of perturbations on scale $R$ is given by the mass variance

$$\sigma^2(R,t) = \frac{1}{2\pi^2} \int W_R^2(k) P(k,t) k^2 \mathrm{d}k\,, \tag{24}$$

where $W_R^2(k)$ is the Fourier transform of the top-hat window function with radius $R$. Observational constraints on the amplitude of perturbations on large scales (from weak lensing, cluster counts and redshift space distortion) are often quoted in terms of $S_8 \equiv \sigma_8(\Omega_{\mathrm{m}}/0.3)^{0.5}$ where $\sigma_8$ is the mass variance at $R = 8\mathrm{h}^{-1}\,\mathrm{Mpc}$ i.e. on cluster scales. A scale $R$ goes non-linear (and structure formation starts) once $\sigma(R,t) \approx 1$. Fig. 3 shows the power spectrum, $P(k)$, and the mass variance, $\sigma(M)$, for hot and cold dark matter.

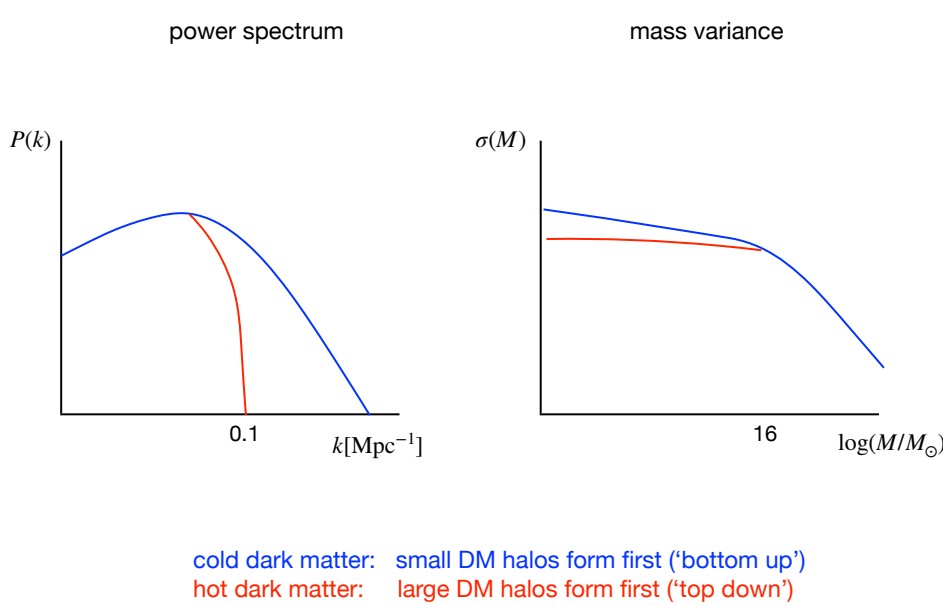

Figure 3: A sketch of the power spectrum, $P(k)$, (left) and mass variance, $\sigma(M)$, (right) for cold dark matter (blue) and hot dark matter (red).

For CDM, the overall shape of the power spectrum depends mainly on $\Omega_{\mathrm{m}}h$, which determines the horizon scale at radiation-matter equality. There are also oscillations in the power spectrum due to the effects of baryons, which depend on the values of $\Omega_{\mathrm{b}}h^2$ and $\Omega_{\mathrm{m}}h^2$. Baryon acoustic oscillations (BAOs) are a 'standard ruler' feature in the clustering of galaxies, which arises from the sound horizon at decoupling (which also sets the scale of the acoustic peaks in the CMB, see Sec. 2.3.2).

We wrap this section up by looking briefly at the spherical collapse model for the formation of individual DM halos, for a more detailed treatment see e.g. Refs. [2, 3]. The radius, $r$, of a spherical overdense region evolves according to the parametric solutions to the Friedmann equation for a closed universe: $r = A(1-\cos\theta)$, $t = B(\theta-\sin\theta)$, where $A$ and $B$ are constants. At early times $r \propto t^{2/3}$ (as for a flat matter dominated universe) and $\delta \propto r$. The expansion subsequently slows down, and at $\theta = \pi$ the region 'turns around' and starts collapsing. Formally $r = 0$ at $\theta = 2\pi$, however the assumptions behind spherical collapse (that matter is in spherical shells with small random velocities) breakdown and the region 'virialises' (i.e it

ends up obeying the virial theorem $2T + V = 0$ where $T$ and $V$ are the kinetic and potential energies). The density of the resulting DM halo is

$$\rho(t_{\mathrm{col}}) = \Delta\,\rho_{\mathrm{c},0}(1 + z_{\mathrm{col}})^3, \tag{25}$$

where 'col' denotes the epoch at which $\theta = 2\pi$, $\Delta$ is the virial overdensity and $\rho_{\mathrm{c},0}$ is the present day critical mass density. For a flat matter dominated universe $\Delta = 18\pi^2 = 178 \approx 200$. There are fitting functions for the redshift dependent $\Delta$ for cosmologies with non-zero cosmological constant or curvature [11], however sometimes $\Delta = 200$ is used for $\Lambda$CDM (see Sec. 3.2.2).

## 1.5   Gravitational lensing

Gravitational lensing occurs when matter deflects/bends the path of light as it travels from a source to the observer. Strong lensing (for a review as of 2010 see Ref. [12]) occurs when the deflection is large and multiple images are formed (or an Einstein ring if the source, lens and observer are aligned). The image properties (i.e. number, positions, fluxes) depend on the mass distribution. For instance substructure in the form of DM subhalos (see Sec. 3.2) can be probed via flux ratios and gravitational imaging (see Sec. 4.1). Microlensing [13] occurs when the angular separation of images is too small to be resolved ($\sim$ micro arc second) and instead the source is temporarily brightened, for a review see Ref. [14]. Microlensing is a particularly powerful technique for probing compact DM, for instance Primordial Black Holes (see Sec. 4.3 and lectures by Bernard Carr and Florian Kühnel). Weak lensing (for a review as of 2015 see Ref. [15]) occurs when the deflection is small. Cosmic shear, the distortion of images of distant galaxies due to weak lensing, allows the matter distribution to be mapped, and $\Omega_{\mathrm{m}}$ to be constrained (see Secs. 2.2 and 2.4).

## 1.6   Inflation

Inflation is a period of accelerated expansion ($\ddot{a} > 0$) in the early Universe. It was proposed to solve several problems with the Big Bang. Namely

- **Flatness**: if the universe isn't exactly flat, the density evolves away from the critical density (for which the geometry is flat), and for the density to be as close to the critical density as it is today (see Sec. 2.3), the density at early times has to be extremely close to the critical density.

- **Horizon**: regions that weren't in causal contact before decoupling have the same CMB temperature and anisotropy distribution.

- **Monopole/massive relic**: if massive particles/topological defects are formed when symmetry breaks they would come to dominate the universe (and prevent successful nucleosynthesis etc.).

Inflation solves these problems, respectively, by driving the initial (post inflation) density extremely close to the critical density, allowing the observable universe to originate from a small region that was in causal contact prior to inflation, and diluting monopoles/massive relics.

Accelerated expansion requires negative pressure, which can be achieved by a scalar field, $\phi$, with a sufficiently flat potential, $V(\phi)$. In the simplest single-field models, inflation ends when the potential becomes too steep, the field then oscillates around the minimum of its potential and decays creating a radiation dominated universe. This is known as reheating. Quantum fluctuations in the scalar field generate density (or scalar) perturbations which are close to scale invariant, and can, depending on the shape of the potential, be consistent with measurements of the temperature anisotropies in the CMB (see Sec. 2.3).

## 1.7 Type 1a supernovae

Type 1a supernovae are explosions that occur when mass accretion from a binary companion leads to a white dwarf exceeding the Chandrasekar mass limit (the maximum mass that can be supported by electron degeneracy pressure). They are standardisable (rather than completely standard) candles; there is an observed correlation between their maximum absolute brightness and the rate at which they fade. This allows them to be used to measure the luminosity distance, and hence constrain the present day matter and cosmological constant density parameters.

The luminosity distance, $d_L$, is the distance calculated for an object of known luminosity, $L$, (a standard candle) assuming that the inverse square law for flux $f$ holds: $d_L = [L/(4\pi f)]^{1/2}$. In reality the expansion of the universe reduces the flux by a factor of $(1+z)^2$ (the energy of individual photons and the rate at which photons arrive are both reduced by $(1+z)$) and the area the photons spread out over is changed if the universe isn't flat. For a flat ($k = 0$) universe, using Eqs. (5) and (11),

$$d_L = r(1+z) = (1+z)\int_{t_e}^{t_0}\frac{\mathrm{d}t}{a(t)} = (1+z)\int_0^z\frac{\mathrm{d}z'}{H(z')} \tag{26}$$

$$\approx \frac{1}{H_0}\left[z + \frac{1}{2}(1-q_0)z^2 + ...\right], \qquad \text{for } z \ll 1, \tag{27}$$

where $t_e$ is the time that light is emitted from an object at redshift $z$, and $q = -\ddot{a}/(aH^2)$ is the deceleration parameter. For a flat universe containing matter and a cosmological constant, the deceleration parameter today is given by $q_0 = \Omega_{m,0}/2 - \Omega_{\Lambda,0}$.

In the late 1990s two teams (the High z Supernovae Search team [16] and the Supernovae Cosmology Project [17]) found that supernovae at redshifts $z > 0.1$ are dimmer than expected for a decelerating matter-dominated universe, and hence the expansion of the Universe is accelerating. A recent analysis of the Pantheon sample, a compilation of $\sim 1000$ type 1a supernovae out to $z \approx 2$, finds $\Omega_{m,0} = 0.30 \pm 0.02$ and $\Omega_{\Lambda,0} = 0.70 \pm 0.02$ [18]. Tighter constraints are obtained when this data is combined with CMB, BAO and local $H_0$ measurements. When the equation of state parameter of dark energy, $w$, is allowed to differ from $-1$, using the combined data sets, $w = -1.01 \pm 0.09$, which is consistent with a cosmological constant. The observations are also consistent with $\mathrm{d}w/\mathrm{d}z = 0$, i.e. no time variation of the equation of state parameter.

## 1.8 Summary of the history of the Universe

Table 1 summarises the history of the Universe. The Universe is radiation dominated at early times, with nucleosynthesis, the synthesis of the nuclei of the light elements, starting $\sim 1\,\mathrm{s}$ after the Big Bang. The matter dominated era starts at $t \sim 0.05\,\mathrm{Myr}$, with the formation of atoms and the 'release' of the CMB occurring shortly after ($t \sim 0.3\,\mathrm{Myr}$). Structures start forming at $\sim 0.1\,\mathrm{Gyr}$, and then $\Lambda$, or dark energy, comes to dominate the Universe, causing its expansion to accelerate, at $t \sim 10\,\mathrm{Gyr}$. The age of the Universe, $t_0$, (i.e. how long after the Big Bang 'today' is) is calculated for a $\Lambda$CDM universe with $\Omega_{m,0} = 0.30$ and $\Omega_{\Lambda,0} = 0.70$.

Table 1: Summary of the history of Universe.

| What | When |
|---|---|
| nucleosynthesis (formation of the nuclei of light elements) | $\sim 1\,\mathrm{s}$ |
| radiation-matter equality | $0.05\,\mathrm{Myr}$ |
| recombination, decoupling and last scattering (atoms form, CMB 'released') | $0.3\,\mathrm{Myr}$ |
| structure formation starts | $\sim 0.1\,\mathrm{Gyr}$ |
| matter-$\Lambda$ equality | $10\,\mathrm{Gyr}$ |
| today | $13.7\,\mathrm{Gyr}$ |

## 2 Evidence for dark matter

### 2.0 Introduction

In this section we'll look at the observational evidence for cold, non-baryonic dark matter, starting with galaxies (Sec. 2.1), before moving out in scale to galaxy clusters (Sec. 2.2), the cosmic microwave background anisotropies (Sec. 2.3) and large scale structure (Sec. 2.4). We conclude the section with a very brief mention of modified gravity (Sec. 2.5).

**Recommended further reading**

The observational evidence for DM is covered in many places, including Bertone, Hooper & Silk [19], chapter 1 of Profumo's book 'An introduction to particle dark matter' [20], and the sections of the 'Particle Data Group review of particle physics' [5] on Dark matter (Baudis & Profumo), the Cosmic microwave background (Scott & Smoot) and Cosmological parameters (Lahav & Liddle). If you're interested in a historical perspective, see Sander's book [21], Bertone & Hooper's review [22] or chapters 6 and 7 of Peebles' 'Cosmology's century: an inside history of our modern understanding of the universe' [23]. Ryden's cosmology textbook covers the evidence for dark matter [1]. Binney & Tremaine's 'Galactic dynamics' textbook [24] is the classic reference for the theory of galaxies, while Bovy is writing an interactive, online graduate textbook [25] on this topic.

### 2.1 Galaxies

Some of the most straightforward and long standing evidence for DM comes from the rotation curves of spiral galaxies [26,27]. In a spiral galaxy stars and gas clouds move in circular orbits due to gravity, and their speeds can be measured using the Doppler shift of the Hydrogen 21cm line.

Using Newton's law of gravity (and also Newton's shell theorem: the gravitational force outside a spherical shell of matter is the same as if all the matter were concentrated at a point at its centre) the rotation, or circular, speed is given by

$$v_{\mathrm{c}} = \sqrt{\frac{GM(<r)}{r}}, \tag{28}$$

where $M(<r) = \int_0^r 4\pi r^2 \rho(r)\,\mathrm{d}r$ is the mass enclosed within a radius $r$, and $\rho(r)$ is referred to as the density profile. Outside of the matter distribution (i.e. at large $r$ where $\rho(r)$ is expected to be zero) then $v_{\mathrm{c}} \propto r^{-1/2}$. This is what is observed for the planets in the Solar System, and

is sometimes referred to as a 'Keplerian fall off'. At large $r$ (greater than the extent of the stellar disc) the rotation speeds are in fact observed to be, roughly, constant. This tells us that $M(< r) \propto r$ and $\rho(r) \propto r^{-2}$, i.e. (if Newtonian gravity is correct) the luminous components of a spiral galaxy are surrounded by an extended invisible dark matter halo.

There are a couple of caveats to note:

- Not all rotation curves are exactly flat, see e.g. Ref. [28].

- Dark matter halos extend to larger radii than the luminous components, and simulated halos have density profiles that are shallower (steeper) than $r^{-2}$ at small (large) radii (see Sec. 3.2.2).

Additional evidence for dark matter comes from the stability of disk galaxies [29]. Self-gravitating disks form bars ('bar instability') unless they have large velocity dispersion. Embedding disks in a massive, extended, roughly spherical halo is a solution to this problem.

## 2.2 Galaxy clusters

Galaxy clusters contain 100s or 1000s of galaxies as well as hot X-ray emitting gas. They are the largest gravitationally bound objects in the Universe, therefore we expect that the material they contain is representative of the Universe as a whole. They provide us with evidence for, and information about, dark matter through three different types of observation.

- **Total mass from the virial theorem** [30,31]

  In a self-gravitating system the kinetic energy ($T$) and potential energy ($V$) are related by the virial theorem: $2T + V = 0$ (see e.g. Ref. [24]). To apply the virial theorem we need to relate the kinetic and potential energies to observable quantities. The mean square velocity can be written as

  $$\langle v^2 \rangle = \frac{\sum_i m_i v_i^2}{\sum_i m_i} = \frac{2T}{M} \, , \tag{29}$$

  where $M = \sum_i m_i$ is the total mass, while the potential energy is given by

  $$V = -\frac{1}{2} \sum_i \sum_{j \neq i} \frac{G m_i m_j}{r_{ij}} \, . \tag{30}$$

  If we define a gravitational radius, $R_{\mathrm{G}}$,

  $$R_{\mathrm{G}} = 2 \left( \sum m_i \right)^2 \left( \sum_i \sum_{j \neq 1} \frac{m_i m_j}{r_{ij}} \right)^{-1} \, , \tag{31}$$

  then $V = -G M^2 / R_{\mathrm{G}}$ and the total mass can be written in terms of the mean square velocity and the gravitational radius as: $M = R_{\mathrm{G}} \langle v^2 \rangle / G$. The mean square velocity can be calculated from the measured (with the Doppler effect) galaxy speeds, while the gravitational radius can be estimated from their projected positions, allowing us to estimate the total mass. This typically gives a mass to luminosity ratio

  $$\frac{M}{L} \sim 400 \frac{M_\odot}{L_\odot} \, , \tag{32}$$

  where $M_\odot$ and $L_\odot$ are the Solar mass and luminosity respectively. This is equivalent, roughly, to a mass density parameter $\Omega_{\mathrm{m}} \sim 0.3$. See e.g. Ref. [1] for more details.

- **Baryon fraction from X-ray gas**

  The baryon fraction is the fraction of the total mass of a galaxy cluster in the form of baryons $f_{\rm b} = M_{\rm b}/M_{\rm tot}$. Assuming that galaxy clusters provide a 'fair sample' of the Universe, the baryon fraction is also equal to the ratio of the baryon and matter density parameters: $f_{\rm b} = \Omega_{\rm b}/\Omega_{\rm m}$.

  Assuming the gas is spherically symmetric and in hydrostatic equilibrium (so that the pressure gradient force and gravity balance):

  $$\frac{1}{\rho}\frac{{\rm d}P}{{\rm d}r} = -\frac{GM(<r)}{r^2}\,. \tag{33}$$

  Using the ideal gas law, $P = k_{\rm B}\rho T/\mu m_{\rm p}$, this can be rewritten as

  $$\frac{k_{\rm B}T}{\mu m_{\rm p}}\left(\frac{{\rm d}\ln T}{{\rm d}\ln r} + \frac{{\rm d}\ln\rho}{{\rm d}\ln r}\right) = -\frac{GM(<r)}{r}\,. \tag{34}$$

  The 1st term in the brackets on the LHS can be measured from X-ray spectra, while the 2nd term can be measured from X-ray surface brightness measurements. The result is an estimate of the baryon fraction $f_{\rm b} \sim 0.144 \pm 0.005$ [32]. There are systematic errors on this value from e.g. deviations from hydrostatic equilibrium and uncertainties in the cluster temperature-mass relation.

- **Mass distribution from gravitational lensing**

  Strong lensing of a background galaxy by a galaxy cluster (e.g. CL0024+1654) produces multiple images of the background galaxy and, from the positions and intensities of these images, the mass distribution within the galaxy cluster can be deduced [33].

  Merging clusters, like the bullet cluster [34], are a particularly interesting special case. In the bullet cluster a smaller subcluster of galaxies has passed through the main cluster. However the hot X-ray emitting gas (which is the dominant baryonic component of clusters) interacts and lags behind, with the gas in the smaller subcluster having a bullet-like shock front. Weak lensing allows the gravitational potential to be reconstructed, and it's found that the total mass is concentrated around the galaxies in the cluster and subcluster. Therefore the clusters must contain a significant amount of non-baryonic DM. The lensing analysis assumes general relativity, however explaining these observations of merging clusters without DM is a major challenge for modified gravity models. We'll see in Sec. 4.2 that merging clusters also allow DM self-interactions to be constrained.

## 2.3 Cosmic microwave background anisotropies

### 2.3.1 Amplitude

As we'll see in Sec. 2.3.2, the detailed scale dependence of the temperature anisotropies in the CMB allow the total, matter and baryon density parameters to be measured precisely. However the measured typical amplitude of the fluctuations, $\Delta T/T \approx 10^{-5}$, alone provides evidence for non-baryonic DM.

The amplitude of the primordial perturbations can be measured from the CMB temperature anisotropies: the density perturbations, $\delta\epsilon$, lead to fluctuations in the gravitational potential, $\nabla^2(\delta\Phi) = 4\pi G\delta\epsilon$, which in turn generate red/blue shifts of photons: ($\delta T/T = \delta\Phi/3$). As outlined in Sec. 1.4, sub-horizon density perturbations in matter grow proportional to the scale factor, $a$, from radiation-matter equality at $t_{\rm eq} \approx 0.05\,{\rm Myr}$. Baryons, however, are tightly coupled to photons until decoupling, at $t_{\rm dec} \approx 0.4\,{\rm Myr}$ and therefore perturbations in baryons

can only grow after decoupling. Consequently in a universe without non-baryonic DM the initial density perturbations have to be larger, and produce larger temperature fluctuations in the CMB, $\Delta T/T \sim 10^{-4}$, for observed structures to form. In other words, non-baryonic DM is required for perturbations to grow sufficiently from the initial amplitude, as measured from the CMB anisotropies.

### 2.3.2 Characteristic angular scale

The positions of the acoustic peaks in the CMB angular power spectrum (Sec. 1.3.2) (or equivalently the typical size of the hot/cold spots) are largely sensitive to the geometry of the Universe, and hence the total energy. The positions are determined by the ratio of the sound horizon (the maximum distance sound waves can have travelled) at last scattering, $d_{\mathrm{hor}}(t_{\mathrm{ls}})$, to the angular diameter distance, $d_{\mathrm{A}}$, the distance an object of known length, i.e. a standard ruler, appears to have:

$$\theta_{\mathrm{hor}} = \frac{d_{\mathrm{hor}}(t_{\mathrm{ls}})}{d_{\mathrm{A}}} . \tag{35}$$

The horizon distance at scattering is given, Eq. (6), by

$$d_{\mathrm{hor}}(t_{\mathrm{ls}}) = a(t_{\mathrm{ls}}) \int_0^{t_{\mathrm{ls}}} \frac{\mathrm{d}t}{a(t)} , \tag{36}$$

while the angular diameter distance for a flat ($k = 0$) universe of an object with extent $l = ar\delta\theta$ is

$$d_{\mathrm{A}} \quad \equiv \quad \frac{l}{\delta\theta} = \frac{ar\delta\theta}{\delta\theta} = \frac{1}{1+z} \int_{t_{\mathrm{ls}}}^{t_0} \frac{\mathrm{d}t}{a(t)} \tag{37}$$

$$\approx \quad \frac{d_{\mathrm{hor}}(t_0)}{z} , \qquad \text{for } z \gg 1 . \tag{38}$$

The measured value of $\theta_{\mathrm{hor}}$ is close to expectations for a flat universe and, from the 2018 Planck temperature, polarisation and lensing data [10],

$$\Omega_k = 1 - (\Omega_{\mathrm{m}} + \Omega_{\Lambda}) = -0.0106 \pm 0.0065 , \tag{39}$$

i.e. the total energy density is very close to the critical density for which the geometry of the universe is flat.

The baryon and matter densities affect the oscillations in the photon fluid and hence the heights of the Doppler peaks. Increasing the baryon density increases the amplitude of the odd peaks, while the height of the 3rd peak is sensitive to the cold dark matter density (see e.g. Wayne Hu's webpages for detailed explanations [35]). From the 2018 Planck temperature, polarisation data and lensing data [10]

$$\begin{aligned} \Omega_{\mathrm{b}}h^2 &= 0.02237 \pm 0.00015 , \\ \Omega_{\mathrm{cdm}}h^2 &= 0.1200 \pm 0.012 . \end{aligned} \tag{40}$$

This precise determination of the baryon density parameter is consistent with the independent, and much higher red-shift, measurement from nucleosynthesis (Sec. 1.3.1).

## 2.4 Large scale structure

Large scale structure observations are typically not as powerful or clean a probe of cosmological parameters on their own as the CMB anisotropies are (galaxies are biased tracers of the matter distribution, redshift is a combination of expansion and peculiar velocity etc.). However,

different observables have different degeneracies (combinations of parameters that they're insensitive to), so combining data sets can lead to more precise constraints (provided that they're consistent). For instance, the Dark Energy Survey finds, from an analysis combining cosmic shear, galaxy clustering and galaxy-galaxy lensing, $\Omega_m = 0.34 \pm 0.03$ [36]. Combined with other cosmological datasets, including Planck, BAO, BBN and $H_0$, the measurement tightens to $\Omega_m = 0.306^{+0.004}_{-0.005}$.

## 2.5 Modified gravity

All of the observational evidence for DM to date comes from its gravitational effects. Therefore it's not unreasonable to ask whether the observations could instead be explained by modifying the laws of gravity. While Newton's laws have been tested to high accuracy on terrestrial scales, the laws of gravity could, in principle, be different on astronomical/cosmological scales. Explaining all of the diverse observations discussed above, on scales ranging from individual galaxies to the Universe as a whole is, however, a major challenge. See Justin Khoury's lectures.

# 3 Dark matter distribution

## 3.0 Introduction

As mentioned immediately above, all of the observational evidence we have for DM comes from its gravitational effects. If we want to confirm the existence of DM (and the standard $\Lambda$CDM cosmological model) we need to detect it. The signatures expected in DM detection experiments depend on how it's distributed. For instance, laboratory based direct detection experiments (see Igor Irstorza's lectures for axions and Jody Cooley's for classical WIMPs) probe the DM density and speed distribution at the Solar radius, $R_\odot = 8.2\,\text{kpc}$ [37], within the Milky Way (MW), while WIMP indirect detection via annihilation products (see Tracy Slatyer's lectures) is sensitive to the DM spatial distribution.

We first look briefly at theoretical modelling of the DM distribution in the MW, including the 'standard halo model' (Sec. 3.1) before moving on to results from numerical simulations (Sec. 3.2) and observations (Sec. 3.3). We conclude the section by looking at potential 'small-scale challenges' (Sec. 3.4). In this section we'll focus on 'vanilla', completely collisionless, CDM. We'll look at how the distributions of warm and self-interacting DM differ in Sec. 4, while Lam Hui's lectures are focused on 'fuzzy' ultralight dark matter.

**Recommended further reading**

The textbooks by Binney & Tremaine [24] and Bovy [25] cover the theory of the dark matter distribution in galaxies in detail. The reviews by Bullock & Boylan-Kolchin [38] Kuhlen, Vogelsberger & Angulo [39], and Zavala & Frenk [40] cover numerical simulations, with the first focussing on small-scale challenges (Sec. 3.4). Note that significant progress has been made, in particular with hydrodynamical simulations which include baryonic physics, since Ref. [39] was written in 2012. Annika Peter's lectures are focussed on numerical simulations. Helmi's recent review [41] covers observations of streams and substructures within the Milky Way.

## 3.1 Theory

The number of particles with phase space coordinates in $\mathbf{x} \to \mathbf{x} + d\mathbf{x}$ and $\mathbf{v} \to \mathbf{v} + d\mathbf{v}$ at time $t$ is given by $f(\mathbf{x}, \mathbf{v}, t)d^3\mathbf{x}d^3\mathbf{v}$, where $f(\mathbf{x}, \mathbf{v}, t)$ is the phase space distribution function. The phase space distribution function, $f$, of a collection of collisionless particles is given by the solution

of the collisionless Boltzmann equation:

$$\frac{\mathrm{d}f}{\mathrm{d}t} = 0.$$ (41)

In Cartesian coordinates this becomes

$$\frac{\partial f}{\partial t} + \mathbf{v}.\frac{\partial f}{\partial \mathbf{x}} - \frac{\partial \Phi}{\partial \mathbf{x}}\frac{\partial f}{\partial \mathbf{v}} = 0,$$ (42)

where $\Phi$ is the potential. In a self-consistent system (where the density distribution generates the potential) the potential and density are related by Poisson's equation:

$$\nabla^2 \Phi = 4\pi G \rho = 4\pi G \int f \, \mathrm{d}^3 \mathbf{v}.$$ (43)

Collisionless particles can change their energy and reach a steady state if they experience a fluctuating gravitational potential (a process known as 'violent relation'). As we saw in Sec. 1.4 structure formation happens hierarchically, and real DM halos haven't reached a steady state. They contain substructure in the form of subhalos and tidal streams (see Secs. 3.2 and 3.3), and the velocity distribution contains imprints of the halo's assembly history.

The standard halo model (SHM) is the simplest model of a DM halo, and is widely used in, e.g., the analysis of data from WIMP direct detection experiments (see Jody Cooley's lectures). It is an isotropic, 'isothermal'[7] sphere with density profile $\rho(r) \propto r^{-2}$. In this case the solution of the collisionless Boltzmann equation is a so-called Maxwellian velocity distribution, given by

$$f(\mathbf{v}) = N \exp\left(-\frac{3|\mathbf{v}|^2}{2\sigma^2}\right),$$ (44)

where $N$ is a normalisation constant. The isothermal sphere has a flat rotation curve and the r.m.s. velocity dispersion is related to the the circular speed (the speed with which objects on circular orbits orbit the Galactic centre), $v_c^2 = r(\mathrm{d}\Phi/\mathrm{d}r)$, by $v_c = \sqrt{2/3}\,\sigma$.

The density distribution of the SHM is formally infinite and hence the velocity distribution extends to infinity too. In reality the Milky Way halo is finite, and particles with speeds greater than the escape speed, $v_{\mathrm{esc}}(r) = \sqrt{2|\Phi(r)|}$, will not be gravitationally bound to the MW. This is often addressed by simply truncating the velocity distribution at the measured local escape speed, $v_{\mathrm{esc}}(R_0)$.

The standard parameter values used for the SHM are a local density $\rho(R_0) = 0.3\,\mathrm{GeV\,cm^{-3}}$, a local circular speed $v_c(R_0) = 220\,\mathrm{km\,s^{-1}}$, and a local escape speed $v_{\mathrm{esc}}(R_0) = (550-600)\,\mathrm{km\,s^{-1}}$. We will discuss the determination of these parameters, including their latest values and uncertainties, in Sec. 3.3.

## 3.2 Numerical simulations

### 3.2.1 Introduction

In CDM cosmologies structure forms hierarchically: small halos (on average) form earlier, and then larger halos form via mergers and accretion (for a visualisation see e.g. Ref. [42]). Subhalos are reduced in mass, or destroyed, by tidal stripping as they orbit within larger halos (see e.g. Ref. [43]).

N-body simulations (e.g. Aquarius [42]) contain only dark matter. In recent years there has been significant progress in hydrodynamic simulations (e.g. APOSTLE [44], Auriga [45],

---

[7]This name arises since the phase-space distribution function has the same form as that of an isothermal self-gravitating sphere of gas.

FIRE [46]) which include baryons (i.e. stars and gas) using prescriptions for 'sub-grid' physics. Baryons affect the DM distribution in various ways, for instance baryonic contraction (the infall of baryons pulls in the DM, steepening the DM density profile) [47], stellar feedback (can reduce the DM density in inner regions of a halo, leading to the formation of a constant density core), and disk shocking (a rapid gravitational perturbation, which increases the internal energy of a subhalo).

A heuristic 'non-expert' summary of the processes involved in simulating a MW-like halo is as follows

- Choose input cosmological parameters (e.g. $h, \Omega_m, \Omega_\Lambda, n_s$) and calculate the input power spectrum of density perturbations.

- Carry out a large volume simulation.[8]

- Select MW-like halo(s): $M \approx 10^{12} M_\odot$, no massive close neighbours (or alternatively one M31-like neighbour) or recent major mergers.

- Resimulate at higher resolution, i.e. using lower mass 'particles'[9] in region that forms halo of interest ("zoom technique").

- Carry out convergence tests e.g. do the properties you're interested in change when you change the 'particle' mass or softening of the gravitational force law?

Here we focus on the aspects of simulated halos that are most relevant for dark matter indirect and direct detection, namely the density profiles of MW-like and dwarf halos (Sec. 3.2.2), subhalo mass function and radial distribution (Sec. 3.2.3), and the local velocity distribution (Sec. 3.2.4). For complete, detailed coverage see Annika Peter's lectures.

### 3.2.2 Density profile of halos

First, some technicalities. Halos are defined via their virial radius $r_{vir}$, the radius within which the density is $\Delta$ times the background density $\bar{\rho}$, where $\Delta$ is the virial overdensity (see Sec. 1.4). The virial mass is then the mass within this radius $M_{vir} = 4\pi r_{vir}^3 \Delta \bar{\rho}/3$. However different authors use different conventions for the value of the virial over-density ($\Delta = 200$, or the redshift dependent fitting function for $\Lambda$CDM [11] which has $\Delta(z = 0) = 333$) and the background density (critical density or matter density), see e.g. Ref. [38,48]. A more practical way to parameterise halo size is to use the maximum circular speed: $v_{max} = \sqrt{GM(<r)/r}|_{max}$.

The density profiles of halos in DM-only simulations are generally well fit by the Navarro-Frenk-White (NFW) profile [49]:

$$\rho(r) = \frac{\rho_0}{(r/r_s)[1 + (r/r_s)]^2}, \tag{45}$$

where the scale radius, $r_s$ is the radius at which the logarithmic derivative of the density profile is equal to $-2$: $(d \ln \rho / d \ln r)_{r=r_s} = -2$. For $r \ll r_s$, $\rho(r) \propto r^{-1}$, while for $r \gg r_s$, $\rho(r) \propto r^{-3}$. However high resolution DM-only simulations find density profiles that deviate from a pure power law as $r \to 0$ (e.g. Ref. [50]), and are better fit by the so-called Einasto profile [51]

$$\rho(r) = \rho_s \exp\left\{-\frac{2}{\alpha}[(r/r_s)^\alpha - 1]\right\}, \tag{46}$$

with shape parameter $\alpha \approx 0.1 - 0.2$ [52]. The Einasto profile has a logarithmic slope equal to $-2(r/r_s)^\alpha$ i.e. it decreases smoothly as $r$ decreases. A quantity that is sometimes useful when describing DM halos is the concentration, $c = r_s/r_{vir}$.

---

[8]This is far more complex than these six words make it sound...

[9]The mass of simulation particles is necessarily much, much greater than the mass of DM particles.

Baryons affect the density profile at small $r$ where their density is highest. The nature of the effect depends on the size of the halo, and on how the baryonic physics is modelled. For instance in the EAGLE hydrodynamical simulations of MW-like halos the density profile is steeper than NFW for $r \sim (1.5-6)$ kpc due to baryonic contraction [53]. Stellar feedback is most efficient at producing cores (constant density inner regions) with radius $r_{\rm core} \sim (1-5)$ kpc in bright dwarf galaxies (e.g. Refs. [54,55]). In some simulations cores with radius $r_{\rm core} \sim (0.5-2)$ kpc form in MW-sized galaxies (whether a core forms depends on the gas density threshold for star formation [55]).

### 3.2.3 Mass function and radial distribution of subhalos

Subhalos in DM only simulations have a power-law mass function $\mathrm{d}n/\mathrm{d}M \propto (M/M_\odot)^{-\alpha}$ with $\alpha = 1.90 \pm 0.03$, e.g. Ref. [50]. Roughly 10% of the halo mass is in resolved subhalos and their distribution is 'anti-biased' i.e. there are less subhalos in the inner regions, where the overall density is higher, as their mass is reduced more effectively there. At the Solar radius $< 0.1\%$ of the total mass is in resolved subhalos. In hydrodynamical simulations the fraction of subhalos is smaller, and is reduced more at small radii, with the size of the reduction depending on how the baryons are modelled, e.g. Ref. [56].

### 3.2.4 Local velocity distribution

Hydrodynamic simulations of MW-like halos find the velocity distribution in the Solar neighbourhood is fairly well fit by a Maxwellian velocity distribution, Eq. (44), [57–59]. The Maxwellian distribution is a better fit to simulations with baryons than to DM-only simulations. This is possibly because baryonic contraction makes the logarithmic slope of the density profile at the Solar radius closer to the $-2$ of the standard halo model. There are, however, potentially significant deviations from a Maxwellian in the tail of the speed distribution. 'Debris flow' are features due to incompletely phased mixed material [60,61], while the Large Magellanic Cloud increases the number of high-speed particles [62].

## 3.3 Observations

### 3.3.1 Introduction

There has been huge progress in understanding the properties and history of the MW in recent years thanks to the Gaia satellite [63]. Gaia is an ongoing ESA space astrometry mission (2013-2022+?), measuring the positions, parallaxes and proper motions (change in apparent position) of $> 1$ billion stars in the MW ($\sim 1\%$ of the total number of stars). Analyses often also use information on metallicity from spectroscopic surveys e.g. APOGEE [64], RAVE [65], LAMOST [66]. It should also be noted that there isn't a rigid divide between theory/simulations and observations; modelling is required to interpret observations and the statistical errors are often now so small that systematic errors are significant, or even dominant.

As in the previous subsection, we will focus on the aspects of the DM distribution that are most relevant for DM detection: the local density and circular speed (Sec. 3.3.2), the density profile of the MW and dwarf galaxies (Sec. 3.3.3), the local escape speed (Sec. 3.3.4) and finally features in the local velocity distribution, in the form of the Gaia-Enceladus/sausage and tidal streams (Sec. 3.3.5). For a complete overview of the status of observations of the MW see Ref. [41], and for details of the implications for DM detection experiments talks by O'Hare [67,68].

### 3.3.2 Local density and circular speed

Various techniques are used to constrain the local DM density, i.e. the DM density at the Solar radius, $\rho(R_\odot)$. These techniques can be divided into two classes: local (using kinematics of nearby stars) and global (e.g. mass modelling). Mass modelling involves using multiple data sets (e.g. rotation curve, velocity dispersions of halo stars, local surface mass density, total mass, ...) to constrain a model (luminous components and DM halo) of the MW. The statistical errors on measurements can be quite small: e.g. $\rho(R_\odot) = (0.36 \pm 0.02)\,\text{GeV}\,\text{cm}^{-3}$ for the best fit NFW halo from a 'unified rotation curve of the MW' combining a large number of data sets [69]. The spread in measurements, however, is much larger than the statistical error on individual measurements: $\rho(R_\odot) \sim (0.3-0.6)\,\text{GeV}\,\text{cm}^{-3}$. This indicates that systematic errors, from e.g. assumptions of equilibrium and spherical symmetry, and modelling uncertainties, are significant. For extensive details see Read's 2014 review [70], and for a more recent review, see de Salas & Widmark [71].

The local circular speed, $v_c(R_\odot) = [r(\text{d}\Phi/\text{d}r)]_{R_\odot}^{1/2}$, can also be determined by multiple methods. For instance, from the proper motion of Sgr A$^\star$, the massive black hole at the MW's centre, $v_c(R_\odot)/R_\odot = (30.3 \pm 0.9)\,\text{km}\,\text{s}^{-1}\,\text{kpc}^{-1}$ [72]. The Solar radius has recently been measured to high precision and accuracy by the GRAVITY collaboration, using the orbit of the star S2 around Sgr A$^\star$: $R_\odot = (8.178 \pm 0.013 \pm 0.022)\,\text{kpc}$ [37]. Using this new value of $R_\odot$ gives $v_c(R_\odot) = (248 \pm 7)\,\text{km}\,\text{s}^{-1}\,\text{kpc}^{-1}$. Another method is Jeans analysis using tracer stars. By taking moments of the collision Boltzmann equation (in cylindrical coordinates)

$$v_c^2(R) = \langle v_\phi^2 \rangle - \langle v_\phi^2 \rangle \left( 1 + \frac{\partial \ln \nu}{\partial \ln R} + \frac{\partial \ln \langle v_R^2 \rangle}{\partial \ln R} \right), \tag{47}$$

where $\nu$ is the density of the tracer stars. For instance combining data from Gaia, APOGEE and other sources: $v_c(R_\odot) = (229.0 \pm 0.2)\,\text{km}\,\text{s}^{-1}$ with a $(2-5)\%$ systematic uncertainty from sources including the uncertainty in the distribution of the tracer stars [73].

### 3.3.3 Density profile of Milky Way and dwarf galaxies

It's hard to measure the inner slope of the MW's density profile, as baryons dominate at small radii. The large measured microlensing optical depth towards the Galactic centre implies large stellar densities in the inner MW, which appears to be in tension with the high DM densities at small radii of 'cuspy' density profiles [74]. Dynamical modelling, taking into account various observational constraints on the Galactic bulge and bar, find a MW DM density profile which flattens to a core or shallow cusp at $r \lesssim 1\,\text{kpc}$ [75].

Dwarf galaxies are a good target for WIMP indirect detection searches via $\gamma$-rays as they are DM dominated, and have high DM densities (see Tracy Slatyer's lectures). Rotation curve measurements find inner slopes that are shallower than NFW, and in some cases consistent with central constant density cores, albeit with significant individual uncertainties and scatter [76].

### 3.3.4 Local escape speed

The escape speed is the speed required to escape the gravitational field of the MW, $v_\text{esc}(r) = \sqrt{2|\Phi(r)|}$. The local escape speed, $v_\text{esc}(R_0)$, is estimated from the speeds of high velocity stars, using a parameterisation of the shape of the high speed tail of the velocity distribution, $f(|\mathbf{v}|) \propto (v_\text{esc}(R_\odot) - |\mathbf{v}|)^k$. Using high velocity stars from the RAVE survey and $2.3 < k < 3.7$ (motivated by numerical simulations): $v_\text{esc}(R_\odot) = 533^{+54}_{-41}\,\text{km}\,\text{s}^{-1}$ [77]. With a similar approach using Gaia data, but without assuming a potential in the modelling: $v_\text{esc}(R_\odot) = 580 \pm 63\,\text{km}\,\text{s}^{-1}$ [78].

### 3.3.5 Features in the local velocity distribution

A significant fraction of the halo near the Sun is in the Gaia-Enceladus/sausage, which is the aftermath of a major merger with a $M \sim 10^8 \, M_\odot$ dwarf galaxy $(8-10)$ Gyr ago [79]. The name arises from the fact the stellar component has radially biased orbits, and hence the distribution of $v_r$ is sausage like [80]. The fraction of the local DM density it comprises is $\sim (10-30)\%$. See Ref. [81] for a refinement of the standard halo model which includes this component.

The local stellar halo also contains narrow tidal streams from smaller or more recent mergers, such as S1 [82], the Helmi streams [83] and Nyx [84]. We'll look at how tidal streams throughout the MW halo can be used to probe the nature of DM in Sec. 4.1.

## 3.4 Small scale challenges

The CDM 'small scale crisis', or probably more accurately (c.f. Ref. [38]) 'small scale challenges', refers to the apparent differences between the results of numerical simulations and observations on sub-galactic scales which first emerged in the 1990s. Briefly the problems are:

- **Cusp-core (or density)**: DM only simulations produce halos with cuspy inner density profiles ($\rho(r) \propto r^{-\gamma}$ with $\gamma \approx 1$) while galaxies, in particular low mass DM dominated dwarfs, have shallower, or even cored ($\gamma \sim 0$) profiles.

- **Missing satellites**: simulated MW-size halos contains thousands of dwarf galaxy sized sub-halos, but 'only' $\sim 50$ dwarf galaxies have been observed (n.b. observations are 'incomplete'; not all of the dwarfs that exist have been observed).

- **'Too-big-to-fail'**: too few medium sized ($M_{\rm dm} \sim 10^{10} M_\odot$) galaxies observed.

As we already saw in Sec. 3.2, these discrepancies could be due to the difficulties of modelling 'sub-grid' baryonic physics. Alternatively they could be resolved by having DM properties or interactions different to 'vanilla' collisionless CDM (see Sec. 4). For a detailed overview see Ref. [38] and Annika Peter's lectures.

# 4 Constraints on the properties of dark matter

## 4.0 Introduction

In this section we focus on how DM properties and particle interactions (i.e. deviations from 'vanilla' collisionless CDM) affect the DM distribution, and hence how these interactions can be probed by astronomical and cosmological observations. As emphasised in Ref. [85], particle interactions can lead to primordial and/or evolutionary deviations from vanilla CDM. Primordial effects modify the evolution of density perturbations, and hence, for instance, the initial halo mass function. Evolutionary DM interactions within halos modify, for instance, their density profiles. We will focus on two widely studied cases: warm DM [86] (Sec. 4.1) and self-interacting DM [87] (Sec. 4.2). Other possibilities include baryon scattering DM (see e.g. Refs. [88] and [85]) and 'fuzzy' DM (see Lam Hui's lectures). We conclude in Sec. 4.3 with a very brief mention of a non-particle dark matter candidate, Primordial Black Holes, which will be covered in detail in Bernard Carr and Florian Kühnel's lectures.

**Recommended further reading**

The LSST[10] dark matter group white paper [88] covers the theory of DM particle interactions, and current (and potential future) constraints. Buckley & Peter review gravitational probes of DM physics [85], while the Bullock & Boylan-Kolchin 'small scale challenges' review [38], includes particle interactions as a potential resolution to these challenges. Tulin & Yu review DM self-interactions and small scale structure [89].

## 4.1  Warm dark matter (WDM)

Warm dark matter (WDM) is semi-relativistic at creation. This is the case for thermally produced DM with mass $m_{\rm wdm} \sim \mathcal{O}({\rm keV})$ [86]. A concrete candidate is the sterile neutrino, for a review see Ref. [90] and Joachim Kopp's lectures. For thermal WDM free streaming erases perturbations and suppresses the power spectrum on mass scales smaller than

$$M_{\rm h} \sim 10^{10} \left( \frac{m_{\rm wdm}}{1\,{\rm keV}} \right)^{-3.33} M_\odot \,, \tag{48}$$

and consequently structure formation is suppressed below this mass. The erasure of perturbations depends on the velocity distribution of the particles. For sterile neutrinos the mass scale beneath which structure formation is suppressed is slightly different from Eq. (48), and depends on the thermal history [91]. WDM halos have cuspy density profiles like CDM, e.g. Ref. [38].

The linear power spectrum, and hence the mass of WDM particles, can be probed via the Lyman-alpha forest [92]. The Lyman-alpha forest is the absorption lines in spectra of galaxies and quasars from the Lyman-alpha electron transition in intergalactic neutral hydrogen 'clouds' (see Sec. 2.1.1 of Ref. [93] or Sec. 2.1.4 of Ref. [94]). The absorption/transmission probability depends on the matter density, leading to limits on the mass of thermal WDM in the range $m_{\rm wdm} > (3-5)\,{\rm keV}$ [95, 96].

The minimum halo mass, $M_{\rm h}$, and hence the WDM particle mass, can be probed by comparing predictions for the abundance of dwarf satellite galaxies in MW-like galaxies with observations [97], taking into account the incompleteness of the observations. The resulting limits are of order $m_{\rm wdm} > 2\,{\rm keV}$ (e.g. Ref. [98]). A DM sub-halo passing by a stellar stream gives the stars in the stream a kick and perturbs their orbits, leading to a gap in the stream which is observable for sub-halo masses $M_{\rm h} \gtrsim 10^5\,M_\odot$ [99]. Analysis of the density power spectrum of the trailing arm of the GD-1 stream finds $m_{\rm wdm} > 4.6\,{\rm keV}$ [100].

Substructure can be probed by strong lensing in two ways. In strong lensing systems with multiple images, substructure can affect one of the images, leading to anomalies in their relative fluxes [101, 102]. From analysis of multiple such systems $m_{\rm wdm} > 5\,{\rm keV}$ [103, 104]. Gravitational imaging looks for substructure induced changes in the shape of lensed emission in long arcs [105], and from analysis of multiple systems $m_{\rm wdm} > 0.2\,{\rm keV}$ [106, 107].

For a compilation of all the constraints on WDM, see Table 4 of Ref. [108].

## 4.2  Self-interacting dark matter (SIDM)

Self-interacting DM (SIDM) is DM particles which scatter elastically with each other [87], for reviews see Refs. [88, 89]. The interaction rate is given by

$$R = n_{\rm dm} \sigma v_{\rm rel} = \frac{\rho_{\rm dm} \sigma v_{\rm rel}}{m} = 0.1\,{\rm Gyr}^{-1} \left( \frac{\rho_{\rm dm}}{0.1\,M_\odot\,{\rm pc}^{-3}} \right) \left( \frac{v_{\rm rel}}{50\,{\rm km}^{-1}\,{\rm s}^{-1}} \right) \left( \frac{\sigma/m}{1\,{\rm cm}^2\,{\rm g}^{-1}} \right), \tag{49}$$

---

[10]LSST is now known as the Vera C. Rubin Observatory.

where $n_{\rm dm}(\rho_{\rm dm})$ is the DM number (mass) density, $\sigma$ the cross section, $v_{\rm rel}$ the relative velocity and $m$ the mass. The mean-free path between interactions, $\lambda = (n\sigma)^{-1} = (\rho\sigma/m)^{-1}$, is of order a kilo-parsec for $\sigma/m \sim (0.1 - 10)\,{\rm cm}^2{\rm g}^{-1}$ and therefore interactions can lead to thermalisation and the formation of a constant density core in the inner regions of DM halos. In the simplest models the sub-halo mass function is the same as for standard CDM, e.g. Ref. [38]. In specific particle physics models (e.g. dark photon) $\sigma/m$ can be velocity dependent (so the effects are different in different mass halos) and couplings with other particles in the early universe can suppress the power spectrum (and hence structure formation) on small scales.

SIDM can be constrained by observations of merging clusters, including the bullet cluster (e.g. Ref. [89] and references therein). DM self-interactions transfer momentum between the cluster halos, so they lag behind the collisionless galaxies, leading to an offset between the DM and galaxies. Merging clusters also lead to constraints from the cluster surviving the merger, and from changes to the mass-to-light ratio from mass loss. There are also constraints from the formation of $\mathcal{O}(10)\,{\rm kpc}$ radius constant density cores in massive galaxies, e.g. Ref. [109], and the ellipticity of halos. CDM halos are triaxial, while DM self-interactions isotropise DM particle velocities, and erase ellipticity at small radii, e.g. Ref. [110], and the shape of DM halos can by probed observationally via cluster strong lensing. The constraints (and potential observations) lie in the range $\sigma/m \sim (0.1 - 1)\,{\rm cm}^2\,{\rm g}^{-1}$, see Table 1 of Ref. [89]

### 4.3 Non-particle dark matter

While the majority of DM candidates are new fundamental particles, DM doesn't have to be a particle. Non-particles candidates, such as Primordial Black Holes (PBHs), can also be constrained by various astrophysical and cosmological processes. PBHs are black holes, with masses $M > 10^{15}\,{\rm g}$, that form in the early Universe from large over-densities. There are constraints on the abundance of PBHs from the consequences of their evaporation, microlensing, gravitational waves from mergers, the consequences of accretion and dynamical effects. See Bernard Carr and Florian Kühnel's lectures and Refs. [111–113].

## 5 Summary

The $\Lambda$CDM cosmological model, in which the Universe is flat with 25% of the energy density being in the form of cold, non-baryonic, dark matter is a good fit to a wide range of observations (nucleosynthesis, CMB, large scale structure, type 1a supernovae). However we don't (yet...) know what dark matter (or dark energy) is. To detect dark matter we need to know how it's distributed, in particular in the Milky Way galaxy, and there has been significant recent (and ongoing) progress in this field from both observations and simulations. We can also probe the nature of dark matter (i.e. deviations from 'vanilla' cold dark matter which just interacts gravitationally) using astronomical and cosmological observations.

## Acknowledgements

AMG is grateful to: the students who attended the School (for the great questions that led to clarifications of various points), Jamie Bolton (for comments and references on Lyman-$\alpha$ forest probes of warm dark matter), Simon Goodwin and Tejas Satheesh (for assistance with proofreading), and Annika Peter (for numerous helpful comments).

**Funding information**

AMG is supported by STFC grant ST/P000703/1.

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
