# Peer review of "Dark Matter in Astrophysics/Cosmology"

_SciPost Physics Lecture Notes, doi:SciPost Phys. Lect. Notes 37 (2022)_

## Round 1 · Referee Report · Anonymous (Referee 1) · 2021-10-15

Strengths

1 - This set of lecture note is well written
2 - It provide a pedagogical explanation of relevant topics

Report

This is a set of lecture notes for an introductory course on Dark Matter and Cosmology. The material covers the basic elements of Cosmology and provides a direct road to further topics involving Dark Matter.

This review is part of a series of lecture notes on Dark Matter and I believe it accomplishes the goal of being accessible for students while working as a background source for the other lecture notes.

As a minor comment, I would suggest to the author to do a quick check on the notation for the energy density in different parts of the manuscript. In some points, the energy density is denoted by $\epsilon$, while in others it is denoted by $\rho$. It would be better to have a uniform notation.

In general, I believe the paper is compatible with the standards of this journal and I recommend it for publication.

---

## Round 1 · Referee Report · Anonymous (Referee 2) · 2021-10-28

Strengths

1- very clearly written and pedagogical introduction to dark matter 2- many references for further reading 3- accessible with no prior knowledge of the subject 4-both historical and modern developments covered

Report

This is an excellent introduction to the topic of of dark matter, well suited for students with little or know prior knowledge of the subject, but also for researchers looking for key references on specific topics.

Requested changes

My comments are only very minor corrections:

  • in Eq (1), Newton's constant isn't introduced
  • a few times \rho appears instead of \epsilon, eg below Eq (2), in the caption and label of Fig 1, in Section 1.4, ...

---

## Round 1 · Referee Report · Anonymous (Referee 3) · 2021-10-30

Report

These lectures notes are supposed to provide introductory material on dark matter in astrophysics and cosmology. The author manages to cover a number of points, including evidence for dark matter, distribution, and constraints.

Including all this material was not a good strategy though, since in my opinion quantity came at the expense of quality. My major criticism is the following: - these notes contain a very comprehensive list of facts, but no explanations. As such, the paper can provide a good summary for someone who already knows the topic, but it would not be helpful for someone who is starting to learn it (like a student). Thus, I would not identify the paper as lecture notes, rather as a review (to be anyhow improved). For this reason, I do not think that the paper can be accepted in the journal SciPost Lecture Notes or, at least, it would require a very dramatic improvement of the text.

Regardless of my decision, and of whether the paper will be published here or in another journal, I would like to provide the author some inputs on minor points to be improved: - The current version of these lectures notes appears to be way too colloquial, and there are many typos. Also, the author uses contracted forms (it's, won't, 'll, 've) that should not be used in proper written English and should be replaced by their full form (it is, will not, will, have). There are also inconsistencies in the notation (use of $\epsilon$ versus $\rho$). - I would suggest that the author, after eq. (1), spells out the word "dot" instead of using the math symbol $\dot$, which in the text appears invisible. - After "Universe is very close to flat" k should not be k=0, but rather $k\approx 0$. - The author says that they will use natural units, but then re-introduces the speed of light $c\neq 1$ in eq. (4). This makes the explanations confusing. - An explanation of redshift is missing. More generally, a lot of concepts are introduced without explanation. - A derivation /explanation of eq. (6) is missing. Same for eq. (14). - Right after eq. (19), it would be really good if the itemized could be accompanied by a figure showing the various regimes. - At the beginning of section 1.4: technically, in the context of GR it is wrong to speak about "force of gravity". Right after that, "a scale" is introduced, but it is not clear what the author is referring to. - A definition of Jeans length is missing. - Close to eq. (23), it would be nice to have an explanation of how the pivot scale is chosen. - The explanation in the subsection on "Modified gravity" seems to be a bit too short, considering how much literature is around. I also think that, from that paragraph, a student could erroneously think that dark matter provides better explanations than modified gravity, while this is not necessarily the case. - "Lab" is too colloquial, I would use the extended word. - Footnote 8: not a very useful comment, it would be more useful to have an explanation of what the 6 words mean.

---

## Round 2 · Author Response

Referee 3 clearly dislikes the scope and style of the lecture notes, however different people can legitimately make different decisions about how to present material. The referee may have been taught that contractions shouldn't be used in written English, however this view isn't universally shared. In writing these notes I've prioritised readability over "proper[LY] written English".
As Sec. 0 explains, the lectures on which these notes are based were designed to "provide the participants in the ‘Les Houches Summer School 2021: Dark Matter’ with the background knowledge of cosmology/astrophysics required for the other courses during the School". As explained in the notes, it would usually take 20+ hours of lectures to cover the material in 'Sec. 1 Introduction to Cosmology' and "therefore our treatment will be necessarily superficial (and unsatisfactory for fans of rigour)" and "This will be a rapid ‘crash course’ for people who haven’t already studied such material and a recap for those who have.". The positive feedback from the participants in the School indicates that they found this approach effective.

---

## Round 2 · List of Changes

In response to Referee 1 and 2's specific comments I have:
i) changed the errant rhos (for energy density) to epsilons (in the text after Eq.(2), in Fig. 1 and its caption, in the 2nd paragraph of Sec. 1.4) and where it's the mass (rather than energy) density that's being referred to I've now explicitly stated this.
ii) Inserted the definition of G after Eq.(1).
Here are my responses to Referee 3's specific comments:
-I would suggest that the author, after eq. (1), spells out the word "dot" instead of using the math symbol \dot, which in the text appears invisible.
Done.
-After "Universe is very close to flat" k should not be k=0, but rather k≈0.
Done.
-The author says that they will use natural units, but then re-introduces the speed of light c≠1 in eq. (4). This makes the explanations confusing.
I have removed the one errant factor of c.
- An explanation of redshift is missing. More generally, a lot of concepts are introduced without explanation.
Redshift is explained ("The expansion of the universe leads to cosmological redshift of the wavelength of photons: lambda proportional to a”.). The omission of a full derivation is deliberate: the benefits of doing it in the lectures wouldn’t have justified the time taken.
- A derivation /explanation of eq. (6) is missing. Same for eq. (14).
I have expanded the explanation of the horizon distance in eq.(6), however this is one of the "incidental, things which I mention briefly, so that if you encounter them you know roughly what they are.”
Eq.(14) is simply, as stated, the ratio of two Maxwell-Boltzmann distributions, with m_{n} ~ m_p. I have added an intermediate step.
-Right after eq. (19), it would be really good if the itemized could be accompanied by a figure showing the various regimes.
Figure added.
-At the beginning of section 1.4: technically, in the context of GR it is wrong to speak about "force of gravity". Right after that, "a scale" is introduced, but it is not clear what the author is referring to.
I have removed “force of”. These generic statements apply to any scale. I have added “on a given scale” to the sentence “Perturbations…”
-A definition of Jeans length is missing.
The Jeans length is defined: “Perturbations will grow if t_pre > t_dyn which is the case if R is greater than the Jeans length, lambda_J ∼ c_s t_dyn ∼ c_s/(G ρbar)1/2 “
- Close to eq. (23), it would be nice to have an explanation of how the pivot scale is chosen.
I’ve added “(which is usually taken to be roughly in the centre of the range of scales probed by the data)”. A more detailed explanation is unlikely to be of interest or benefit to the target audience of these notes/lectures.
- The explanation in the subsection on "Modified gravity" seems to be a bit too short, considering how much literature is around. I also think that, from that paragraph, a student could erroneously think that dark matter provides better explanations than modified gravity, while this is not necessarily the case.
This subsection is intentionally short because, as mentioned, there were a separate set of lectures covering this topic by Justin Khoury. I firmly believe that this paragraph, and in particular the statement “Explaining all of the diverse observations… is [] a major challenge” is a well-balanced summary of the current state of play. I suspect that many dark matter researchers either wouldn’t have included this subsection, or would have been explicitly negative about modified gravity.
- "Lab" is too colloquial, I would use the extended word.
“lab based” is commonly used and understood terminology in the field of direct detection, however I’ve changed lab to laboratory.
-Footnote 8: not a very useful comment, it would be more useful to have an explanation of what the 6 words mean.
Footnote 8 is a light-hearted comment, acknowledging how demanding carrying out these cosmological simulations is. As mentioned earlier in the notes, there were a separate set of lectures by Annika Peter covering numerical simulation in detail, therefore it would have been undesirable to cover this point in detail in either the lectures or notes.
Following feedback from a student who read the notes after submission, I have also added a few words of explanation before the expression for the deceleration parameter today at the top of page 13.
I have also corrected various minor typos.

---

## Editorial Decision

published